# Pathological Circulating Factors in Moyamoya Disease

**DOI:** 10.3390/ijms22041696

**Published:** 2021-02-08

**Authors:** Yao-Ching Fang, Ling-Fei Wei, Chaur-Jong Hu, Yong-Kwang Tu

**Affiliations:** 1Taipei Neuroscience Institute, Taipei Medical University, Taipei 11031, Taiwan; eugene_88@tmu.edu.tw (Y.-C.F.); b101106018@tmu.edu.tw (L.-F.W.); 2Department of Neurology, Shuang Ho Hospital, Taipei Medical University, New Taipei City 23561, Taiwan

**Keywords:** Moyamoya disease, cerebrovascular disease, collateral formations, RNF213, growth factors, circulating factor

## Abstract

Moyamoya disease (MMD) is a cerebrovascular disease that presents with vascular stenosis and a hazy network of collateral formations in angiography. However, the detailed pathogenic pathway remains unknown. Studies have indicated that in addition to variations in the of genetic factor *RNF213*, unusual circulating angiogenetic factors observed in patients with MMD may play a critical role in producing “Moyamoya vessels”. Circulating angiogenetic factors, such as growth factors, vascular progenitor cells, cytokines, inflammatory factors, and other circulating proteins, could promote intimal hyperplasia in vessels and excessive collateral formation with defect structures through endothelial hyperplasia, smooth muscle migration, and atypical neovascularization. This study summarizes the hypothesized pathophysiology of how these circulating factors affect MMD and the interactive modulation between them.

## 1. Introduction

Moyamoya disease (MMD) is a cerebrovascular disease associated with progressive vascular stenosis, occlusion of the internal carotid artery (ICA), and the formation of excessive, hazy, proximal collateral vessels, which are termed “Moyamoya vessels” [1]. These fragile, smoke-like vessels provide insufficient cerebral perfusion, causing various symptoms, such as ischemic or hemorrhagic stroke, seizure, cognitive impairment, and disability or death in both children and adults with MMD [2,3].

MMD has mostly been identified in East Asia, and the annual incidence is particularly high in Japan, China, Taiwan, and Korea [4,5]. This regional and ethnic characteristic was determined to be strongly associated with genetic factors [6]. Polymorphism of *R4810K* in the RING (encoded by the *Really Interesting New Gene 1*) [7] finger protein *RNF213* was proposed to be the strongest susceptible gene of MMD and mediate the protein-protein interactions and have ubiquitin-protein ligase activity [8,9]. However, not all patients with MMD have the *RNF213* variant, which indicates that the pathology of MMD is a complex pathway that includes genetic factors, environmental factors, and an innate angiogenetic capacity [10,11].

Understanding of the pathology of MMD is limited. Therefore, this review presents abnormal circulating factors that have been identified in patients with MMD, their interactions, and the effects on the pathogenetic neovascularization pathway and vascular characteristics of MMD.

## 2. Growth Factors

Increased or abnormal activity of various growth factors promotes intimal hyperplasia and smooth muscle cell (SMC) migration in vessels, which may be involved in the pathological angiogenesis and vasculogenesis of MMD. Some growth factors that may cause excessive and aberrant angiogenesis in MMD are presented, such as vascular endothelial growth factor (VEGF), basic fibroblast growth factor (bFGF), hepatocyte growth factor (HGF), and platelet-derived growth factor (PDGF-BB) (Table 1). 

VEGF, the only growth factor that has been demonstrated to play a crucial role in vessel formation, was determined to be significantly increased in the plasma and dura mater of patients with MMD [12,13]. High expression of VEGF increases the recruitment of vascular progenitor cells and promotes migration, proliferation, and neovascularization, resulting in pathological collateral vessel formation. Another study reported that the morphological differences in VEGF may be related to postoperative collateral vessel formation in pediatric MMD. The pediatric MMD population less frequently expresses *VEGF*-634CC and exhibits better postoperative collateral vessel formation. Patients with the VEGF-634G allele have poor collateral formation [14]. The angiogenetic activities of VEGF are also affected by the expression of VEGF receptors (VEGFRs). The expression of VEGFR-2, which is the receptor responsible for endothelial development and vessel production [38], was found to be reduced in patients with MMD with more favorable collateral vessel formation. These results indicated that increased VEGFR-2 may facilitate abnormal vessel formation; thus, blocking of VEGFR-2 could be a therapeutic target [15,38].

The signaling protein bFGF increases the motility of endothelial cells and increases VEGF expression by activating the hypoxia-inducible factor (HIF)-α pathway, thereby promoting the proliferation of mesoderm, neuroectoderm-derived cells, and SMCs, which may be involved in stenosis and the occlusion of the ICA [39]. Furthermore, higher cerebrospinal fluid (CSF) levels of bFGF were observed in patients with MMD, and bFGF levels increased after neovascularization from indirect revascularization [16]. These findings indicate that bFGF could be involved in the pathological pathway in MMD and could be a predictor of the efficacy of surgery [17]. However, whether bFGF is a pathogenic factor or a consequence of the hypoxia condition caused by defective MMD vessels remains uncertain [40].

HGF is a strong angiogenetic inducer, mediated by vascular endothelial cells, that induces endothelial cell proliferation and SMC migration [41]. One study reported that highly increased levels of HGF were observed in the CSF, intima, and media of the carotid fork in patients with MMD. Therefore, HGF may be involved in the etiology of MMD [18].

PDGF-AA and PDGF-BB are two types of PDGF with different homodimeric peptide chains that stimulate DNA synthesis and SMC migration in angiogenesis. One study reported that the plasma levels of PDGF-BB were significantly higher in patients with MMD [12], whereas another reported that PDGF-AA and PDGF-BB obtained from the SMCs of patients with MMD exhibited increased cell migration activity, but not DNA synthesis [40], indicating that PDGF may be involved in the mechanism of intimal hyperplasia in MMD through the stimulation of vascular progenitor cells to differentiate into an SMC lineage.

Upregulated growth factors with excessive activities, abnormal morphology, or different responses to specific receptors may activate the recruitment of vascular progenitor cells, stimulate angiogenesis, and even induce the expression of other growth factors, which underlies the pathogenesis of MMD.

## 3. Circulating Progenitor Cells

Circulating progenitor cells, which were found to be overexpressed in MMD, mostly express CD34^+^, CD133^+^, and VEGFR2^+^. Circulating progenitor cells are recruited and modulated by the aforementioned growth factors (especially VEGF). These circulating progenitors, such as endothelial progenitor cells (EPCs) and smooth muscle progenitor cells (SPCs), assist vasculogenesis and vascular remodeling, and may thus be involved in the pathogenesis of MMD [42] (Table 1).

Studies have demonstrated that among EPCs and endothelial colony-forming cells (ECFCs), one subtype of EPCs has characteristics that are more specific to endothelial function than the other subtypes [43]. This subtype may play a critical role in producing pathogenic MMD vessels. One study reported that circulating CD34^+^ cell levels were higher in patients with MMD who had ICA–middle cerebral artery (MCA) stenosis or occlusion, suggesting that the increased circulation of ECFCs is related to the development of pathological neovascularization in the ischemic brain [22]. Increased EPCs in MMD may also induce arteriogenesis and angiogenesis by promoting resting endothelial cell activity [21]. Aside from the aberrant levels of EPCs in the peripheral blood, the alteration ratio between different subtypes of EPCs causes ineffective vessel formation. In pediatric MMD, reduced tube formation and an increased number of senescent-like phenotypes of ECFCs were identified in the peripheral blood, causing delayed reparation of ischemic vessels and abnormal differentiating activities [20]. Similarly, in adult patients with MMD, EPC–CFUs (Colony-forming units), which are related to tube formation activities, and outgrowth cells, which express regenerative response and vasogenic activity in defect tissue, were determined to be reduced before MMD and increased in early MMD, respectively. Moreover, the paracrine function of EPCs supports endothelial stem cells better than the progenitor cells, which are more mature and well-differentiated, contributing to the formation of a fragile, immature MMD vascular network [24]. Delayed reparation of the defect vessels was also determined to be correlated with mitochondrial morphological and functional abnormalities in ECFCs, which may cause occlusion in MMD [23]. Moreover, gene polymorphism in ECFCs was determined to affect tube formation activity. Knockdown of *RALDH2* mRNA in isolated pediatric MMD ECFCs reduces the expression of retinoic acid, thereby downregulating typical capillary formation in vitro [25].

In addition to well-discussed EPCs that are related to tube formation and reparation, the proliferation of SPCs causes intimal thickening, which is a prominent histological feature in MMD. Studies have indicated that the atypical activities of SPCs may be correlated with their gene expression. Because multiple genes express differentially, SPCs have an irregular morphology, and may produce abnormal arrangements and thickened tubules in MMD [27]. Mutation of the *ACTA2* gene also contributes to increased proliferation in SMCs, causing occlusive disease [26].

Hyperplasia of these ineffective subgroups of progenitor cells may result from gene polymorphism or mitochondrial abnormalities, which induce poor or atypical vessel formation with thickened intima. These findings elucidate the pathological mechanism of stenosis and occlusion in fragile MMD vessels.

## 4. Angiogenesis-Related Cytokines

A variety of circulating cytokines cooperate with growth factors to promote neovascularization, steno-occlusive change, and collateral vessel formation in Moyamoya pathogenesis [44] (Table 1).

Transforming growth factor-β1 (TGF-β1) is a hypothesized enhancer of pathological arteriogenesis in MMD. The levels of TGF-β1 in SMCs derived from the superficial temporal artery and serum were determined to be higher in patients with MMD [28]. Furthermore, the levels of elastin mRNA expression induced by TGF-β1 were increased in SMCs [19]. These two results suggest that the overexpression of TGF-β1 in SMCs or induced by inflammation causes an increase in the expression of connective tissue genes, which promotes elastin synthesis and angiogenesis. TGF-β1 expression could also be caused by defective circulating Treg cells that lack typical suppressive functions and have been determined to be increased in the peripheral blood of patients with MMD [29].

Retinoid acid (RA) plays a critical role in regulating normal angiogenesis by attenuating the neointimal formation and proliferation of SMCs. However, higher expression of cellular RA-binding protein-1 (CRABP-1), which has been reported to downregulate RA, has been identified in the CSF of pediatric patients with MMD [30]. One study indicated that CRABP-1 attenuated RA activity by promoting the production of RA-metabolizing enzymes and accelerating the degradation rate [45], which may enhance atypical vessel formation in patients with MMD.

Matrix metalloproteinase-9 (MMP-9) is an angiopoietin closely related to the bioavailability of VEGF. MMP-9 modulates vasculogenic and angiogenic processes by targeting collagen IV and the other extracellular matrix. MMP-9 enhances the activity of gelatinase, causing the degradation and remodeling of collagen IV, which may be followed by pathological angiogenesis in MMD [31]. Other studies have also reported elevated serum and plasma MMP-9 levels in patients with MMD, indicating that the upregulation of MMP-9 may promote intimal hyperplasia and excessive collateral vessel formation. Therefore, the defective structure of vessels could be caused by hemorrhage in MMD [12,32].

Hypoxia-inducible factor (HIF)-α in the intima of MCA samples and serum monocytes chemoattractant protein-1 (MCP-1) was also higher in patients with MMD. HIF-1 may induce intima thickening by activating the transcription of genes involved in vascular regulation, such as the VEGF and TGF-β families [33]; MCP-1 facilitates the migration of stromal cells and causes pathological collateral vessel formation [12].

These cytokines were suspected to be involved in intimal hyperplasia and excessive collateral vessel formation with defective and fragile structures, through the modulation of ECM (extracellular matrix, e.g., elastin synthesis and digestion of collagen IV) or attenuation of regular angiogenesis. However, whether these cytokines are causative of or merely related to the pathophysiology of MMD, or whether they precede its development, remains unclear [10].

## 5. Inflammatory and Immune Mediators

Inflammatory responses promote intimal hyperplasia and neovascularization, causing stenosis/vessel occlusion and further collateral formation, both of which are observed in MMD. These relative proinflammatory and anti-inflammatory cytokines are mediated by immune cells, such as M2 macrophage and Treg cells, indicating that the pathology of MMD may be related to the immune-mediated inflammatory pathway [46] (Table 1).

One study reported that the levels of sCD163 (marker of M2 macrophage) and CXCL5 (cytokine related to the severity of autoimmune reactions) were higher in the serum of patients with MMD. This finding indicates that M2 macrophages may increase autoimmune activity and may be involved in the pathogenesis of MMD [35]. Elevated levels of IL-1β, secreted by macrophages, were also identified in patients with MMD, and may activate the proliferation of macrophages, endothelial cells, and SMCs, resulting in increased vascular permeability and endothelial dysfunction [12]. Furthermore, the abnormal deposition of antibodies due to inflammation also destroys vessel structure. Excessive immunoglobulin G (IgG) and S100A4 proteins were identified in the intracranial vascular walls of patients with MMD, indicating that the deposition of IgG causes damage to the internal elastic lamina and causes S100A4 + SMC migration, which causes lumen stenosis and compensatory vessel formation [34].

Although the dysfunctional intima and vessel stenosis caused by inflammation and immune activities appear to coincide with the characteristics observed in MMD, some studies have defined MMD as a noninflammatory disease, because no inflammatory cells or macrophage infiltration in the subintimal layer are identified. However, even if inflammatory responses do not cause the pathogenesis of MMD directly, they still affect *RNF213* and promote curative angiogenesis [46].

## 6. Other Factors

Studies have reported that caveolin-1 (Cav-1) is significantly reduced in patients with MMD, especially those with a variant of the *RNF213* gene. Cav-1 may downregulate typical angiogenesis in endothelial cells and promote apoptosis in SMCs, inducing negative arterial remodeling and impairing angiogenesis in MMD [10,36] (Table 1).

NO, modulated by Cav-1, is an angiogenetic factor downstream of VEGF. Endothelial NO synthase (eNOS) and VEGFR-2, which are colocalized in caveolae, are repressed by Cav-1, reducing NO production and tube formation in mice [47]. Reductions in Cav-1 may cause the loss of the suppressed function of eNOS, thus promoting excessive angiogenesis by producing abundant NO. Levels of NO metabolites (nitrate and nitrite) were also observed to be increased in the CSF of patients with MMD and decreased after bypass surgery. Because of the chronic ischemic circulation state, NO metabolites were upregulated and promoted Moyamoya vessel development by dilating collateral vessels and increasing collateral circulation [37]. However, no differences in plasma NO metabolite levels or eNOS polymorphisms were identified between patients with MMD and healthy controls, suggesting that the relationship between NO and MMD pathogenesis warrants further investigation [10,48] (Table 1).

## 7. Discussion

The pathologic pathways of MMD may be strongly correlated with excessive levels and activities of growth factors, causing the recruitment of vascular progenitor cells and thus abundant but defective collateral formation (Figure 1). Cytokines and immune activities, which could result from inflammation and an ischemic environment, were also determined to be involved in the angiogenetic process in MMD.

Among all the growth factors, VEGF seems to play a key role in vascular formation and interacts with each of the other angiogenetic factors in MMD [12,13]. As the factor directly modulating the activities of EPCs and SMCs, VEGF promotes the proliferation and migration of vascular endothelial cells and SMCs, causing neovascularization and intima hyperplasia [38]. The morphological differences among and expression of VEGF receptors are both related to pathological angiogenesis [14,15,38]. Furthermore, VEGF levels are upregulated by TGF-β1 and HIF-1α, which have higher levels of expression in patients with MMD [44]. VEGFR2 and eNOS are suppressed by Cav-1, the levels of which are lower in patients with MMD [37]. These findings all emphasize that the modulation of VEGF through various pathways may be crucial in preventing excessive neovascularization in MMD.

Other growth factors that were determined to be increased in patients with MMD, such as bFGF, HGF, and PDGF, also promote endothelial hyperplasia, SMC migration, and collateral formation: bFGF induces the HIF-α pathway and upregulates VEGF indirectly [15,32,39]; PDGF-BB stimulates vascular progenitor cells to differentiate into an SMC lineage, which strengthens intima hyperplasia [12].

Cerebral inflammation and ischemia states trigger the expression of various factors, including angiogenetic cytokines, immune complex deposits, and other circulating factors, which may cause intimal hyperplasia and compensatory collateral formation. TGF-β and MMP-9 help regulate the extracellular matrix of the vessel wall; once elevated, their activities respectively contribute to an imbalance in elastin and collagen IV levels, resulting in pathological neovascularization with defective vessel structure [12,17,23,24,27,29]. In addition to directly causing endothelial dysfunction by increasing vascular permeability, IL-1β promotes the proliferative activity in SMCs, causing intimal occlusion and compensatory collateral formation [12]. Other deposited cytokines or immune complexes, such as IgG and MCP-1, also exhibit similar responses by facilitating the migration of SMCs [12,28].

Retinoic acid is a protective factor of typical vascular formation that attenuates the proliferation of SMCs. Knockdown of *RALDH2* mRNA in MMD ECFCs also reduces RA levels and could be related to the pathogenesis of MMD [22,25,36].

## 8. Limitations

These circulating factors were determined to be significantly related to MMD. However, whether these factors are causative of MMD or the results of compensatory collateral formation or inflammation and ischemic responses remains unclear.

Furthermore, a strongly suspected pathological factor is gene mutation. The RING finger protein RNF213, which has been identified in familial cases and displays a strong ethnicity effect, is a frequently discussed susceptibility gene for MMD. The presence of the *RNF213* p.4810K variant causes different subtypes of MMD, such as ischemic type and hemorrhagic type [3]. Findings on angiogenetic factors do not explain the differences in subtypes or onset age groups in MMD. Therefore, we still need to figure out whether there is a correlation between genetic variation and abnormal angiogenetic factors.

Recent studies indicate that serum micro-RNAs (miRNAs) associated with *RNF213* mutation are aberrant in MMD. miRNAs are non-coding RNAs responsible for regulating gene transcription. Mediating the angiogenesis and integrity of endothelial cells, specific circulating miRNA may be involved in the pathological signaling pathway. One study found that RNF213-associated miRNA levels were dysregulated in the serum of MMD patients, such as upregulated miR-106b, miR-130a, and miR-126, and downregulated miR-125a-3p, which is related to the ordinary regulation of VEGF and matrix metalloproteinase 11 (MMP11) [49]. Similarly, hsa-miR-6722-3p/-328-3p, interacting with endothelial cell by having an impact on STAT3 (signal transducer and activator of transcription 3,) IGF-1 (insulin Like Growth Factor 1), and PTEN (phosphatase and tensin homolog)-signaling pathways, were abnormally upregulated in an MMD-discordant, monozygotic twin cohort. [50] The EV (vesicle encapsulated) miRNA may also take part in the pathology of MMD by affecting the signaling pathway of angiogenesis [51]. Regulating endothelial function and angiogenesis, the specific circulating miRNA aberrant in MMD patients is not only a potential therapeutic target and diagnostic biomarker, but also a key to better understanding a thorough pathological mechanism in MMD.

Therefore, further research investigating the detailed mechanism of the effect of upstream gene phenotypes on these angiogenetic factors is required.

## 9. Conclusions

Overall, the detailed interactions and regulation between angiogenetic growth factors, vascular progenitor cells, relative circulating factors, and immune cytokines appear to be strongly correlated with endothelial proliferation, SMC migration, and excessive but defective neovascularization, which may cause intimal thickening and occlusion and further compensatory, atypical collateral formation, which are specific features of MMD. However, no conclusive evidence has been presented that can explain the accurate pathogenetic mechanism in MMD, and further studies, such as animal models or gene research of RNF213-associated miRNA, are still needed to thoroughly investigate this complex pathway.

## Figures and Tables

**Figure 1 ijms-22-01696-f001:**
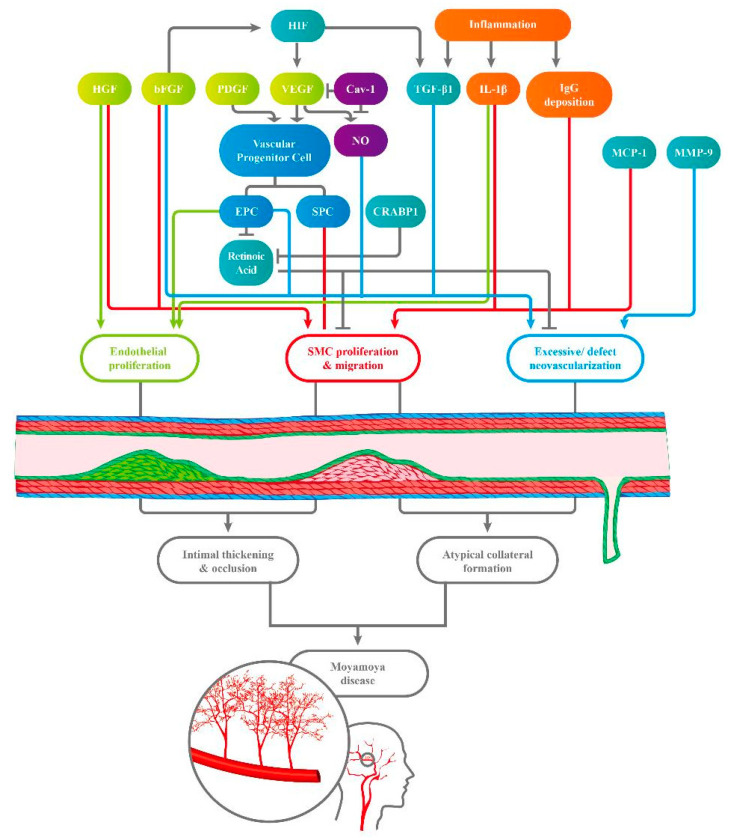
Various circulating factors interact, engendering endothelial proliferation, smooth muscle cell (SMC) proliferation, and migration. These effects result in excessive but defective neovascularization, which causes intimal thickening and occlusion and compensatory atypical collateral formation, both of which are characteristics of Moyamoya disease (MMD). HIF, hypoxia-inducible factor; HGF, hepatocyte growth factor; bFGF, basic fibroblast growth factor; PDGF, platelet-derived growth factor; VEGF, vascular endothelial growth factor; Cav-1, caveolin-1; TGF-β1, transforming growth factor; IL-1β, interleukin-1β; IgG, immunoglobin G; EPC, endothelial progenitor cell; SPC, smooth muscle progenitor cell; CRABP-1, cellular retinoic acid binding protein; MCP-1, monocytes chemoattractant protein-1; MMP-9, matrix metalloproteinase 9.).

**Table 1 ijms-22-01696-t001:** Abnormal findings of circulating factors in patients with Moyamoya disease (MMD) and their hypothesized pathological mechanisms. (↑, upregulated; ↓, downregulated; →, leading to or be related to the consequence in the table. sVEGF1-R, soluble vascular endothelial growth factor 1 receptor; CSF, cerebrospinal fluid; bFGF, basic fibroblast growth factor; HGF, hepatocyte growth factor; PDGF-BB, platelet-derived growth factor-BB; ECFCs, endothelial colony-forming cells; EPCs, endothelial progenitor cells; MCA, middle cerebral artery; CFU, colony-forming unit; *RALDH2*, *retinaldehyde dehydrogenase 2*; *ACTA2*, *actin alpha 2;* IgG, Immunoglobin G; IL-1β, interleukin-1β.).

	Substance	Sampled Population/Sampled Objects	Abnormal Findings in MMD Patients	Hypothetic Pathogenetic Pathway	First Author
Growth Factor	VEGF	MMD patients/ plasma	VEGF ↑	VEGF ↑ → recruitment of vascular progenitor cells and collateral vessel formation	Kang et al. [12]
		MMD patients/ Dura mater	VEGF ↑	VEGF ↑ → migration, proliferation, and neovascularization in vascular cells	Sakamoto et al. [13]
		Pediatric MMD/ gene	*VEGF*−634CC ↓ → better post-operative collateral vessel formation	*VEGF* −634G allele → poor collateral vessel formation (in pediatric MMD)	Park et al. [14]
		MMD patients/ serum	sVEGF1-R and sVEGF2-R ↓ → better collateral vessel formation	1. VEGF2-R → initial steps in angiogenesis 2. VEGF1-R and VEGF2-R↓ → pathological angiogenesis↓	He et al. [15]
	bFGF	MMD patients/ CSF	(bFGF) ↑	bFGF ↑ → pathogenesis in MMD	Takahashi et al. [16]
		MMD patients/ CSF	bFGF ↑ (after neovascularization)	bFGF ↑ → proliferation of mesoderm, neuroectoderm-derived cells, and SMC ↑ → stenosis and occlusion of ICA	Yoshimoto et al. [17]
	HGF	MMD patients/ CSF	(HGF) ↑ (in CSF, the intima and the media of the carotid fork)	HGF (strong angiogenic inducer) ↑ → proliferation of endothelial cells↑ and migration of SMCs ↑	Nanba et al. [18]
	PDGF-BB	MMD patients/ plasma	PDGF-BB↑ PDGF receptors↓ (on SMCs)	PDGF-BB ↑ → vascular progenitor cells differentiating into a SMC lineage → intima hyperplasia	Kang et al. [12]
		MMD patients/ SMC strains	PDGF-AA & PDGF-BB → SMC migration ↑	PDGF ↑ → intima thickening ↑	Yamamoto et al. [19]
Circulating progenitor Cells	ECFCs/ EPCs	Pediatric MMD/ Peripheral blood	tube formation-type ECFCs ↓ senescent-like phenotype ECFCs ↑	EPCs ↓ → delayed repairing in ischemia vessels and abnormal differentiating activities → ineffective vasculogenesis	Kim et al. [20]
		MMD patients/ Peripheral blood	ECFCs ↑	EPCs ↑ → activity of resting endothelial cells ↑ → arteriogenesis and angiogenesis ↑	Rafat et al. [21]
		Patients with ICA or MCA stenosis or occlusion/ peripheral blood	Circulating CD34 + cells ↑	Circulating ECFCs ↑ → pathological neo-vascularization in ischemia brain	Yoshihara et al. [22]
		MMD patients/ peripheral blood	The mitochondria in ECFCs display morphological and functional abnormalities.	ECFCs with abnormal function → delayed repair of the defect vessels → vessel occlusion	Choi et al. [23]
		Adult MMD/ CFU and outgrowth cell	1. CFU↓ (in advanced MMD) 2. EOC↑ (in early MMD)	Inefficient subtype of EPCs → vascular occlusion and angiogenesis	Jung et al. [24]
		Pediatric MMD/ ECFCs in vitro and in vivo	*RALDH2* gene in ECFCs↓ → capillary formation↓	*RALDH2* ↓ → RA ↓ → defective tube formation activity	Lee et al. [25]
	SPCs	MMD patients/ peripheral blood	Mutation in *ACTA2* gene → proliferation in SMCs ↑	SMCs ↑ → occlusive disease	Guo et al. [26]
		MMD patients/ peripheral blood	SMCs make an irregular arrangement and thickened tubules	Multiple genes express differentially → SMCs shows irregular morphology	Kang et al. [27]
Angiogenesis-related cytokines	TGF-β1	MMD patients/ SMCs from superficial temporal artery and serum	TGF-β1 ↑ (in SMCs and in serum)	TGF-β1 ↑ → connective tissues genes ↑ → promoting neovascularization	Hojo et al. [28]
		MMD patients/ SMCs	TGF-β1 ↑ → *elastin* mRNA ↑	TGF-β1 ↑ (produced by MMD SMCs and in inflammatory stimulation) → elastin synthesis ↑	Yamamoto et al. [19]
		MMD patients/ Peripheral blood	Fr III Treg cells ↑ (lack of suppressive functions)	TGF-β↑ (induced by circulating Treg) → VEGF ↑	Weng et al. [29]
	CRABP-1	Pediatric MMD/ CSF	CRABP-1 ↑	CRABP-1 ↑→ retinoid acid ↓ → SMCs migration & proliferation ↑ (induced by growth factors)	Kim et al. [30]
	MMPs	MMD patients/ Serum	autocrine activities of MMP-9 ↑	MMP-9 ↑→ gelatinase↑ → collagen IV↓ and vascular remodeling → angiogenesis ↑	Blecharz-Lang et al. [31]
		MMD patients/ Peripheral blood	Plasma MMP-9 ↑	MMP-9↑ → intimal hyperplasia and excessive collateral vessel formation	Kang et al. [12]
		MMD patients/ Serum	Serum MMP ↑	MMP-9↑ → pathological angiogenesis and defect vascular structure → hemorrhage in MMD	Fujimura et al. [32]
	HIF-1α	MMD patients/ MCA sample	HIF-1α ↑ (in the intima and endothelium)	HIF-1α ↑ → transcription of other growth factors and cytokines ↑ → intima proliferation ↑	Takagi et al. [33]
	MCP-1	MMD patients/ Serum	MCP-1↑	MCP-1 ↑ → the migration of stromal cells↑ → pathological collateral vessel formation	Kang et al. [12]
Inflammatory mediator	IgG Immune complex	MMD patients/ Intracranial vessels	IgG and S100A4 protein ↑ (in vascular wall)	IgG immune complex deposition ↑ → damage in internal elastic intima → S10A4 + SMC migrates into intima → stenosis and occlusion in vessels.	Lin et al. [34]
	M2 macrophage	MMD patients/ Serum	CD163 + M2-polarized macrophage and CXCL5 levels ↑	M2 macrophage ↑ → autoimmune activities ↑	Fujimura et al. [35]
	IL-1β	MMD patients/ Peripheral blood	IL-1β ↑	IL-1β ↑ → macrophage, endothelial cells and SMCs ↑ → vascular permeability ↑&endothelial dysfunction ↑	Kang et al. [12]
Others	Caveolin-1	MMD patients/ serum	Serum Cav-1 ↓	Cav-1↑ → typical angiogenesis ↓ and apoptosis in SMC↑ → impaired angiogenesis	Chung et al. [36]
		MMD patients/ ?	Caveolin-1 ↓ (especially in *RNF213* variant carriers)	Cav-1 is a crucial mediator in MMD	Bang et al. [10]
	NO metabolites	MMD patients/ CSF	1. NO metabolites levels ↑ (in CSF) 2. NO metabolites ↓ after the bypass surgery	Defect collateral vessel formation → the NO metabolites ↑	Noda et al. [37]

## Data Availability

No new data were created or analyzed in this study. Data sharing is not applicable to this article.

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
