# Peer review of "Pathological Circulating Factors in Moyamoya Disease"

_ijms, 2021, doi:10.3390/ijms22041696_

Round 1

Reviewer 1 Report

The authors summarized previous studies regarding several kinds of circulating factors, including growth factors, progenitor cells, and cytokines, in MMD. The description of the various factors which may contribute to the pathogenesis is well organized.

Here are some comments.

  • Involvement of circulating miRNAs are also reported recently in of MMD. Some differentially expressed miRNAs may be related to inflammation or angiogenesis. Please review and discuss them in the manuscript.
  • The authors should cite the below original paper in the introduction.

Suzuki J, Takaku A. Cerebrovascular ‘moyamoya’ disease. Disease showing abnormal net-like vessels in base of brain. Arch Neurol. 1969; 20:288–299.

  • References of previous papers are not sufficient in the introduction and discussion sections.
  • Figure and Tables are not cited in the manuscript.

Author Response

Thank you very much for taking your time to provide professional and valuable advice.

  • Involvement of circulating miRNAs are also reported recently in of MMD. Some differentially expressed miRNAs may be related to inflammation or angiogenesis. Please review and discuss them in the manuscript.

Response 1: The relevant content is added in the “limitation’’ part, marked in red.

  • The authors should cite the below original paper in the introduction.

Suzuki J, Takaku A. Cerebrovascular ‘moyamoya’ disease. Disease showing abnormal net-like vessels in base of brain. Arch Neurol. 1969; 20:288–299.

Response 2: The paper is cited in the introduction.

  • References of previous papers are not sufficient in the introduction and discussion sections.

Response 3: More references are added in the introduction and discussion after referring to other review papers.

  • Figure and Tables are not cited in the manuscript.

Response 4: Figure 1 is cited in the “discussion” part; table 1 is respectively cited in the beginning in every paragraph discussed about the circulating factors.

*The figure is replaced by a new one with same content but different conceptual image of the vascular change in MMD.

Reviewer 2 Report

I have read this original, interesting and also comprehensive review which brings into attention the MMD and its pathophysiology, stressing on the significance of the interactive modulation between circulating angiogenetic factors (growth factors, vascular progenitor cells, cytokines, inflammatory factors), that could stimulate intimal hyperplasia in vessels and extreme collateral formation with flaw structures through endothelial hyperplasia, smooth muscle migration, and atypical neovascularization. This article not only emphasizes the importance of genetic factors, but also presents the abnormal circulating factors that have been identified in patients with MMD, their relations, and the effects on the pathogenesis neovascularization pathway and vascular features of MMD, relying on different studies results.   

From my point of view, this article raises a great interest in further research on investigating the detailed mechanism of the effects of upstream gene phenotypes on these angiogenetic factors; taking into consideration that no conclusive evidence has been offered, additional studies (such as animal models or gene research) are still required to investigate this complex pathway.

I have no comments or suggestions for the authors.

Author Response

Thank you very much for taking the time to provide professional and valuable advice. Thank you very much for your very positive comments on our article.